# Outcome after Prenatal Diagnosis of Trisomy 13, 18, and 21 in Fetuses with Congenital Heart Disease

**DOI:** 10.3390/life12081223

**Published:** 2022-08-12

**Authors:** Stephanie Springer, Eva Karner, Christof Worda, Maria Magdalena Grabner, Elisabeth Seidl-Mlczoch, Franco Laccone, Jürgen Neesen, Anke Scharrer, Barbara Ulm

**Affiliations:** 1Department of Obstetrics and Gynecology, Division of Obstetrics and Feto-Maternal Medicine, Medical University of Vienna, 1090 Vienna, Austria; 2Department of Internal Medicine, Saint Josef Hospital, 1130 Vienna, Austria; 3Department of Pediatric and Adolescent Medicine, Division for Pediatric Cardiology, Medical University of Vienna, 1090 Vienna, Austria; 4Institute of Medical Genetics, Medical University of Vienna, 1090 Vienna, Austria; 5Department of Pathology, Medical University of Vienna, 1090 Vienna, Austria

**Keywords:** congenital heart defect, trisomy 13, trisomy 18, trisomy 21, chromosomal abnormalities, trisomies, prenatal diagnosis, fetal outcome, intrauterine fetal death, perinatal mortality, neonatal mortality

## Abstract

Fetal congenital heart disease (CHD) is often associated with chromosomal abnormalities. Our primary aim was to assess stillbirth and neonatal mortality rates for pregnancies complicated by trisomies 13, 18, and 21 in the presence of CHD, from a single tertiary referral center during 2000–2020 in a retrospective cohort study. The secondary aims were to investigate maternal morbidity in these pregnancies, and to study the gestational or neonatal age when mortality occurred. Inclusion criteria were the prenatal diagnosis of at least one structural CHD, together with prenatally diagnosed fetal trisomy 13, 18, or 21. One-hundred and sixty patients with fetal trisomy 13 (14.4%), fetal trisomy 18 (28.8%), and fetal trisomy 21 (56.9%) were evaluated. In total, 98 (61.3%) families opted for the termination of pregnancy (TOP). Of the remaining 62 (38.8%) pregnancies, 16 (25.8%) resulted in intrauterine fetal death/death during delivery. Ten out of twenty-one (47.6%) infants with trisomy 13 or 18 were born alive. The livebirth rate was 87.8% (36/41) for infants with trisomy 21. Early neonatal death was observed in nine (19.6%) infants. Thirty-one (86.1%) infants with trisomy 21 survived the first year of life. These data may be helpful for counseling affected parents when the decision to terminate or continue the pregnancy should be considered.

## 1. Introduction

Congenital heart disease (CHD) is one of the most prevalently occurring malformations causing neonatal and infant mortality [1]. According to the European Surveillance of Congenital Anomalies (EUROCAT) registry, CHD affects 8 out of every 1000 births, including terminations of pregnancy for fetal anomalies and fetal deaths [2]. 

Epidemiological studies have postulated that genetic or environmental causes are detected in up to 30% of cases [3], whereas gross chromosomal aneuploidies were found only in 8–10% [4]. Besides the increased risk of infant morbidity and mortality, fetuses with CHD carry an increased risk of intrauterine fetal death (IUFD). Several studies report IUFD rates varying from 1 to 38% in fetuses diagnosed with CHD [5,6,7,8,9,10,11,12].

Along with trisomy 21 (Down syndrome), trisomy 18 (Edwards syndrome) and trisomy 13 (Patau syndrome) are the most frequent autosomal aneuploidies [13]. Although these aneuploidies are related to a high rate of spontaneous abortion, IUFD, and a shorter life expectancy, they belong to disorders which are compatible with life [14]. Over the last decades, the early prenatal diagnosis of these common trisomies has improved tremendously due to sophisticated first-trimester ultrasound screening programs and non-invasive prenatal testing of cell-free DNA in maternal blood. 

In a decision-making process about continuing or terminating the pregnancy, questions regarding the natural outcome and survival of the newborn are the focus. Accurate information about the possible course may help parents in this decision-making process.

Our primary aim was to assess the stillbirth and early and late neonatal mortality rates for pregnancies complicated by common chromosomal aneuploidies (trisomies 13, 18, and 21) in the presence of CHD. The secondary aims were to investigate maternal morbidity in these pregnancies, when the parents had decided to continue the pregnancy in the presence of fetal CHD and aneuploidy, and to study the gestational or neonatal age when mortality occurred.

## 2. Materials and Methods

This single center retrospective cohort study was conducted at the Department of Obstetrics and Gynecology, Division of Obstetrics and Feto-maternal Medicine, a tertiary referral center for fetal medicine and CHD at the Medical University of Vienna, Austria. The study was conducted according to the guidelines of the Declaration of Helsinki, and was approved by the Ethics Committee of Medical University of Vienna (1085/2020). Before the ultrasound examinations, all patients signed a consent form that the data may be used for retrospective studies. Medical data from all consecutive patients who were admitted to our center from 2000 to 2020 for fetal echocardiography and whose fetuses had prenatal diagnosis of a structural CHD were extracted from our fetal CHD database. Inclusion criteria for this study were the prenatal diagnosis of at least one structural CHD, together with prenatally diagnosed fetal trisomy 13, 18, or 21. Eighteen fetuses from a previous analysis of our group were also included [15]. With the exception of pregnancies lost to follow-up, all cardiac diagnoses were based on a combination of pre- and postnatal cardiac assessment, with postnatal confirmation either by echocardiography, cardiac surgery, or autopsy. We intentionally did not include other aneuploidies or cases with abnormal results from chromosome micro-array analysis or exome sequencing, as these more detailed genetic analyses were not available during the entire study period, and later, were not universally applied in all pregnancies. 

Maternal and pregnancy characteristics were derived from the obstetric and fetal CHD databases. Extracardiac anomalies were recorded when they were detected prenatally by targeted ultrasound examination, following fetal echocardiography according to ISUOG guidelines [16]. Routine prenatal genetic testing was offered in all cases with a prenatal diagnosis of CHD. In all pregnancies with prenatally diagnosed CHD and confirmed trisomy 13 or 18, parents were offered to continue or terminate the pregnancy. When continuation of the pregnancy was chosen, compassionate care with or without perinatal heart rate surveillance was discussed with the parents. The termination of pregnancy (TOP) was also offered when fetuses had a CHD together with trisomy 21, until 24 weeks’ gestation. After 24 weeks, TOP for trisomy 21 was only offered when the CHD was considered serious, requiring several surgeries with a poor prognosis postnatally, or in the presence of severe additional, extracardiac anomalies. Clinical records were evaluated for data on pregnancy and neonatal outcome, such as gestational age at birth, birth weight, pre- and perinatal mortality, and mortality until one year after live birth.

Gestational age was based on first-trimester ultrasound screening, whenever available, or on the first dating scan recorded. Stillbirth was defined as fetal death after 19 + 6 gestational weeks (GW), and early neonatal mortality as death within 7 days after live birth [17]. Late neonatal death was defined as death between 7 days and the end of the first year after live birth. Gestational age at birth was categorized into term (>/ = 37 GW), preterm (28 to <37 GW), and extremely preterm (<28 GW). 

Statistical analysis was performed with SPSS version 20 for Windows (IBM SPSS Inc., Armonk, New York, NY, USA), and reported as mean (±standard deviation) for normally distributed continuous variables and median (interquartile range) for non-normally distributed continuous variables. A Kolmogorov–Smirnov Test was used to identify non-normally distributed continuous variables. In the case of continuous variables, the two groups were compared using Student’s t-test or the Mann–Whitney U test, as appropriate. Categorical variables were analyzed with the Chi-squared test or Fisher´s exact test. To calculate correlations, Pearson’s correlation was used. A logistic binary regression model was used to test the statistical significance and to evaluate the relationship between different variables. For these multivariate analyses’ regression coefficients beta, their standard errors and Wald test are given. Differences were considered statistically significant if *p*-values ≤ 0.05.

## 3. Results

Out of 1018 fetuses diagnosed with CHD, 169 (16.6%) matched the inclusion criteria. Nine pregnancies were lost to follow-up. Thus, 160 patients were evaluated in this analysis (Figure 1). Twenty-three (14.4%) had prenatally diagnosed trisomy 13, 46 (28.8%) had trisomy 18, and 91 (56.9%) had trisomy 21. The mean gestational age at diagnosis of CHD was 20.5 weeks in fetuses with trisomy 13, 22.8 weeks in fetuses with trisomy 18, and 21.0 weeks in fetuses with trisomy 21 (Table 1). 

Fetuses with trisomy 13 were most likely to have a complex CHD (30.4%), a double-outlet right ventricle (DORV) (17.4%), or tetralogy of Fallot (17.4%), whereas fetuses with trisomy 18 were most likely to have an isolated ventricular septal defect (VSD) (41.3%), a coarctation of the aorta (30.4%), or a DORV (23.9%). The most common CHD in fetuses with trisomy 21 was the atrioventricular septal defect (AVSD) (61.5%), followed by isolated VSD (17.6%) and coarctation of the aorta (16.5%). The distribution of CHDs according to the type of aneuploidy is given in Table 2. 

Extracardiac anomalies (ECA) were diagnosed prenatally in 97 (60.6%) pregnancies. ECAs were most common (95.7%) in fetuses with trisomy 13 (22/23 fetuses) and trisomy 18 (44/46 fetuses). In contrast, only 31 of 91 fetuses (34.1%) with trisomy 21 had ECAs. Additionally, in 64 (40.0%) cases, sonographic soft markers were found (trisomy 13: eight fetuses (34.8%), trisomy 18: 29 fetuses (63.0%), and trisomy 21: 27 fetuses (29.7%)). The detailed distribution of ECAs and soft markers is listed in Table 3. Intrauterine growth restriction (IUGR) at the time of CHD diagnosis was present in 10/23 fetuses with trisomy 13 (43.5%), 31/46 (67.4%) with trisomy 18, and in 14/91 (15.4%) fetuses with CHD and trisomy 21.

In 10/23 (43.5%) of pregnancies with trisomy 13, 14/46 (30.4%) of pregnancies with trisomy 18, and 51/91 (56.0%) of pregnancies with trisomy 21, the results of a detailed sonographic first-trimester screening were available. In the remaining patients, either no first-trimester screening had been performed or the information was missing. Of the pregnancies with sonographic first-trimester screening results, 9/10 were abnormal in fetuses with trisomy 13 (90.0%), 9/14 with trisomy 18 (64.3%), and 42/51 (82.4%) with trisomy 21.

In total, 98/160 (61.3%) families opted for the termination of pregnancy (TOP). TOP was performed in 69.6% (16/23) of fetuses with CHD and trisomy 13, in 69.6% (32/46) with trisomy 18, and in 54.9% (50/91) with CHD and trisomy 21 (Table 1). Univariate analysis showed that the body mass index (BMI) and the gestational age of CHD diagnosis correlated with the decision of TOP. Women that opted for TOP had significantly lower BMI (*p* = 0.044), and CHD was diagnosed significantly earlier (*p* < 0.001). Furthermore, the diagnosis of a hypoplastic left heart syndrome (HLHS) correlated with the decision of TOP (Pearson correlation 0.152, *p* = 0.05). No correlation existed between the decision to terminate the pregnancy and the type of trisomy, the presence of ECAs or IUGR, maternal parity, or whether a first-trimester screening had been performed. TOP rates did not substantially change between 2000–2010 (65.6%) and 2010–2020 (58.6%, *p* = 0.41). Multivariate analysis demonstrated that only the gestational age of diagnosis of CHD influenced the decision of TOP significantly (beta= −0.23, S.E. = 0.056, Wald = 17.2, *p* < 0.001). Tabular information for the univariate and multivariate analysis are not presented.

Maternal complication rates were unchanged in the presence of CHD and fetal trisomies (Table 4).

In 62/160 (38.8%) pregnancies, the parents had the intention to continue the pregnancy. Fifteen out of sixty-two (24.2%) pregnancies ended in spontaneous intrauterine fetal death (IUFD), and one fetus with trisomy 18 died during delivery. IUFD/death during delivery occurred in two (28.6%) fetuses with trisomy 13, nine (64.3%) with trisomy 18, and five (12.2%) with trisomy 21 (Table 1). The gestational age at IUFD was 13 and 14 completed weeks’ gestation in the fetuses with trisomy 13, one with a complex CHD and hydrops, the other with a complex CHD and holoprosencephaly, hygroma, and hydrothorax. Early IUFD at 14 weeks affected one pregnancy with trisomy 18 and AVSD; one fetus died spontaneously at 26 weeks with a VSD, aortic hypoplasia, omphalocele, and ascites. The other IUFDs with fetal trisomy 18 and CHD occurred at gestational weeks 36 and 37, and two at 39, 40, and 41. Of those with trisomy 21 and CHD, the gestational age at IUFD was 15 weeks with hydrops; 23 weeks with severely hydropic placenta; 32 weeks with fetal bradycardia (<75 bpm) and massive myocardial hypertrophy; 34 weeks with VSD and hepatomegaly; and 36 weeks with AVSD, cardiac decompensation, and hydrops. The presence of ECAs (Pearson correlation 0.171, *p* = 0.03), especially hydrops fetalis (Pearson correlation 0.524, *p* < 0.01), the presence of a complex CHD (Pearson correlation 0.323, *p* = 0.01), and the type of trisomy (Pearson correlation −0.286, *p* = 0.02) correlated with the occurrence of IUFD. No correlation was found between the occurrence of IUFD and maternal BMI, maternal age, smoking status, nulliparity, IUGR at diagnosis, and fetal sex.

All five newborns with trisomy 13 that were born alive died within the first week of life (Appendix A). Five (35.7%) infants with trisomy 18 were born alive. Of these, three (60.0%) died within the first week of life; one was followed up until the end of the first week of life, and was then admitted to a palliative care unit outside of our center (with an unknown outcome); one girl with trisomy 18 and atretic ileum, ASD, VSD, and cleft palate, born with a birthweight of 2800 g (10th percentile) by cesarean section, underwent several surgeries, including resection of a coarctation of the aorta, not diagnosed antenatally, died at 5 months.

Thirty-six (87.8%) infants with trisomy 21 and CHD were born alive. One (2.8%) boy with AVSD and tetralogy of Fallot died three hours postnatally, with a birthweight of 4230 g (95th percentile) and maternal insulin-dependent diabetes with poor compliance, and an emergency cesarean section for non-reassuring fetal heartbeats at admission to the delivery ward at 41 + 0 weeks, with an Apgar score of 0/1/1. The autopsy confirmed the cardiac findings, and revealed additional bilateral lung hypoplasia, disseminated pneumonia, hepatosplenomegaly, and liver necrosis. Four children with CHD and trisomy 21 died between 7 days and one year after livebirth (Appendix A). A total of 31/36 (86.1%) infants with CHD and trisomy 21 survived the first year of life. The birth characteristics of livebirths are given in Table 5.

Neonatal survival correlated significantly with the type of trisomy (Pearson correlation 0.596, *p* < 0.01) and the gestational age at delivery (Pearson correlation 0.280, *p* = 0.029). No correlation was found between neonatal survival and maternal age, maternal BMI, nulliparity, smoking status, ECAs and IUGR at diagnosis, and fetal sex.

## 4. Discussion

Of 160 pregnancies with fetal CHD and fetal trisomy 13, 18, or 21, 98 parents (61.3%) opted for the termination of pregnancy. In the remaining 62 pregnancies (38.8%), with the parental choice to continue the pregnancy, stillbirth rates ranged from 64.3% in fetuses with trisomy 18 and CHD, 28.6% in those with trisomy 13, to 12.2% in fetuses with trisomy 21. Spontaneous IUFD correlated with the presence of extracardiac anomalies, fetal hydrops, and the type of trisomy. In fetuses with trisomy 18, most IUFDs occurred near term. The gestational age at IUFD in fetuses with trisomy 21 and CHD varied from 15 to 36 weeks, and was evenly distributed throughout gestation. Early neonatal mortality rates were 100% for liveborn neonates with CHD and trisomy 13, 60% for those with trisomy 18, and 2.8% for neonates with CHD and trisomy 21. Late neonatal death occurred in two children with trisomy 18, and four children with trisomy 21 and CHD. Maternal morbidity during pregnancy and delivery was not affected by the presence of fetal CHD and common trisomies.

First-trimester screening programs, including ultrasound and the analysis of cell-free fetal DNA from the maternal blood, have increased the rates of early fetal aneuploidy diagnoses and early terminations of pregnancy for fetal aneuploidy [1,2]. Still, not all parents choose to undergo early screening, among other reasons, because they are unaware of the specific benefits and risks, they cannot afford the extra costs, or they decline screening for possible aneuploidy. Fetal echocardiography in cases with suspected CHD throughout gestation, thus, examines very different fetal populations, from those who have previously undergone sonographic and genetic screening with normal or abnormal results, to those with little or no previous structural and genetic anomaly screenings. Therefore, data on aneuploidy rates and outcomes in fetuses with prenatally-diagnosed CHD from studies performed before first-trimester screening was widely introduced may not be valid anymore [3]. In the present series of fetuses with CHD, 16.6% had either trisomy 13, 18, or 21, and only approximately half of the pregnancies had undergone first-trimester screening. More than half of the pregnancies studied were fetuses presenting with Down syndrome. As expected, AVSD was the predominant CHD among fetuses with trisomy 21, whereas isolated VSD was most common in fetuses with trisomy 18 [3].

The rate of spontaneous IUFD in pregnancies with known CHD ranged from 12.2% among fetuses with trisomy 21 to 64.3% among those with trisomy 18. The reported stillbirth rates for CHD without genetic anomaly range from 0.2% for simple VSDs, 3.3% for AVSD, to 4.8% for HLHS [4,5]. After the exclusion of TOP, the risk of stillbirth in fetal trisomy 13 averages 22 to 42%, and in fetal trisomy 18, 32 to 63% [4,6] without taking into account the presence of a CHD. Stillbirth rates, as well as peri- and postnatal mortality, however, strongly reflect the level of medical support chosen by the parents and the medical team [7]. In the present series, most fetuses with CHD and trisomy 18 who died spontaneously before birth were near term; all families had chosen ´natural outcome accepting and including a high risk of IUFD´ for the pregnancy and comfort care in cases of live birth. Parents who elect to continue the pregnancy increasingly ask for different levels of medical intervention postnatally, eventually with the goal of prolonging life [7]. When (longer) survival, including high levels of medical support, is what the parents demand, IUFD near term may be ´prevented´ by the early induction of labor or caesarean section, thus resulting in decreased reported stillbirth rates. Birthweight < 1000 g, very low gestational age at birth, trisomy 13, and the goals of care (comfort care) have been associated with lower postnatal survival, whereas male fetal sex and hydrops correlated with higher rates of IUFD [7].

In our collective, 95.7% of fetuses with trisomies 13 and 18 had extracardiac anomalies. The most common extracardiac anomalies comply with those described in the literature, including malformations of the nervous system, facial dysmorphias, limb anomalies, and others [6,8]. In contrast, in 34.1% of fetuses with CHD and trisomy 21, extracardiac anomalies were noted at the time of fetal echocardiography. This may explain why 23.1% of cases with trisomy 21 and CHD in the present series were diagnosed after 24 completed weeks’ gestation.

Parent counseling in the presence of a CHD and fetal trisomy ideally involves several disciplines, including fetal cardiac specialists, geneticists, neonatologists, and psychologists. Appropriate information on the spontaneous risks of fetal or perinatal death is crucial when reflecting options to terminate or continue the pregnancy. When continuation is chosen by the parents, maternal risks that might be specifically linked to fetal disease gain importance, and questions arise on how to monitor both maternal and fetal wellbeing. Our data, although limited by rather small case numbers, especially for fetal trisomies 13 and 18, demonstrate no evidence of increased maternal risks, neither during pregnancy nor perinatally.

Fetal and intrapartum monitoring largely depend on parental choices for the perinatal period. Whether IUFD is an ´accepted´ outcome is a decision the parents may chose after counseling, or whether frequent ultrasound and Doppler measurements during the last trimester should try to prevent this outcome, specifically in trisomies 13 and 18. Forty percent of parents who had decided to continue the pregnancy after the diagnosis of trisomy 13 or 18 and CHD in this study elected caesarean section for delivery, and so did 55.6% of parents with fetal trisomy 21 and CHD.

Only infants with trisomy 21 survived the neonatal period. Early neonatal deaths were observed in all five live births with trisomy 13, in 60% with trisomy 18, and one infant with trisomy 21 died within the first week of delivery. According to the previously published literature, around 40% of infants with trisomy 13 and 18 die within the first week after delivery, whereas half do not survive the first 24 h [4,9,10]. Groen et al. report late neonatal mortality within 28 days after delivery with 13.2% for trisomy 13 and 10.2% for trisomy 18, whereas Meyer et al. indicate 25.5% for trisomy 13 and 37.1% for trisomy 18. Takahashi et al. describe a total neonatal mortality rate of 62% for infants with trisomy 13. For infants with trisomy 13, a median survival time of 2–10 days, and for trisomy 18, 3.5–29 days, were reported regardless of the presence of CHD [6,7,9,10,11,12]. At our neonatal center, infants with trisomy 13 and 18 prenatally diagnosed with CHD were provided with comfort care after delivery, whereas Cortezzo et al. report different neonatal care goals [7]. Parents choose whether intensive care, non-invasive care, or comfort care should be conducted, with the aim that neonatal intensive care can extend survival time in infants with trisomy 13 and 18, but is associated with surgical interventions and morbidity [7,11,12]. In a select group of patients with trisomy 13 or 18, cardiac interventions including the surgical correction of cardiac defects can be performed with acceptable early postprocedural survival rates [13]. A recent analysis of trends in invasive procedures, surgery, and mortality among neonatal hospitalizations with trisomy 13 and 18 in the United States revealed that though gastrointestinal and central nervous system procedures increased from 2009–2010 through 2017–2018, cardiovascular procedures and invasive mechanical ventilation rates decreased in these patients, with no significant changes in mortality or length of survival [14]. Guidelines and recommendations about the termination of pregnancy, management of pregnancies with trisomies, parent information strategies, and neonate support reflect the legal and ethical standards within the respective culture, and vary largely among countries [15,16]. Options to choose from different available interventions and treatments—including the option not to intervene and treat neonates with trisomy 13 or 18—may help parents and medical care teams formulate individualized care plans for these infants [14]. Compared to other common trisomies, overall survival in infants with trisomy 21 is considerably higher [4,17,18,19,20]. Rankin et al. indicate a one-year-survival rate of 82.7% in infants with trisomy 21 and cardiovascular diseases [20].

Because patient selection was based on a fetal CHD registry, fetuses without prenatally-detected CHD were not included. Most studies investigating the perinatal outcomes of fetuses with aneuploidy do not specify on the presence of CHD. Common trisomies are highly associated with cardiovascular diseases [21,22,23]. Infants with Down syndrome are likely to present with AVSD [5,20,21,24], whereas fetuses with trisomy 13 and trisomy 18 are more often diagnosed with isolated VSD and complex heart defects [7,9]. Cortezzo et al. report variable outcomes depending on the type of CHD and approach to intervention [7]. Although fetuses with trisomy 13 and trisomy 18 are more likely to develop complex cardiac defects, studies do not show increased mortality in fetuses with CHD [5,6,9,11,12,25]. In contrast, in fetuses with Down syndrome, an association between the presence of CHD and higher peri- and neonatal mortality has been described [20,26]. A large register-based study from Denmark indicates CHD as the main cause of death in people with Down syndrome in their first two life decades [27].

## 5. Strengths and Limitations

The data analyzed for this study were derived from a large fetal CHD database from a single tertiary referral center with highly consistent clinical management in pregnancies affected by fetal CHD and aneuploidy during the entire study period. Information on the course of pregnancy and delivery, including maternal morbidity, was, thus, available in most cases. Details on whether active or expectant management was chosen for delivery, and on compassionate care, could be studied; these data are often not present in larger studies from registries summarizing the data of several centers. Limitations originate mainly from referral bias and from the inclusion criteria we selected thereof. Fetuses with other, non-cardiac structural anomalies and/or intrauterine growth restriction are more likely to be referred to a center for detailed (malformation and cardiac) evaluation and, eventually, genetic testing, and they may, therefore, be over-represented in the cohort studied. It is likely that we missed several fetuses with either small, isolated, or undetected CHD and aneuploidy, since even experienced fetal echocardiography cannot detect all CHD prenatally. All parents who chose to continue the pregnancy in the presence of trisomy 13 and 18 and CHD in the present series opted for compassionate care after delivery; therefore, no conclusions could be drawn on neonatal survival in these infants if intensive care had been conducted. Further limitations result from the retrospective study design, and, thus, the loss of follow-up data from some pregnancies.

## 6. Conclusions

The outcome after the prenatal diagnosis of CHD in the presence of a common fetal trisomy depends largely on the type of trisomy, and also on the decision of the parents and their supporting medical team on the continuation or termination of pregnancy. Fetuses with CHD and trisomy 13 or 18 may survive until term delivery, with a high risk of fetal mortality at the end of the third trimester for fetuses with trisomy 18, and high neonatal mortality rates. Fetal trisomy 21 and CHD are associated with higher IUFD rates throughout gestation. Extracardiac anomalies, fetal hydrops, and the type of trisomy correlate with mortality rates. Maternal complication rates during pregnancy and delivery, after prenatal diagnosis of a CHD and fetal trisomy, are unaffected by the presence of these fetal anomalies.

## Figures and Tables

**Figure 1 life-12-01223-f001:**
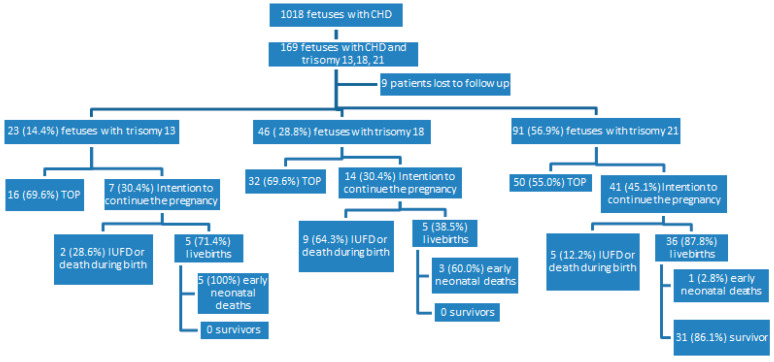
Outcome after prenatal diagnosis of trisomy 13, 18, and 21 in fetuses with congenital heart disease. CHD, congenital heart disease; TOP, termination of pregnancy; IUFD, intrauterine fetal death.

**Table 1 life-12-01223-t001:** Characteristics of 160 pregnancies with prenatal diagnosis of fetal congenital heart disease (CHD) and common trisomies.

Variable	T13 (*n* = 23)	T18 (*n* = 46)	T21 (*n* = 91)
Nulliparity ^1^	11 (47.8)	24 (52.2)	31 (34.1)
Maternal age (years) ^2^	33 (27–39)	36 (31–39)	37 (31–39)
Body Mass Index ^2^	24.3 (21.5–27.1)	22.1 (20.5–25.9)	23.8 (21.7–28.7)
Gestational age at diagnosis of CHD ^3^	20.5 (5.8)	22.8 (6.4)	21.0 (5.7)
Gestational age at diagnosis of CHD ^1^			
<16 + 0 weeks	6 (26.1)	6 (13.0)	23 (25.3)
16 + 0 to 24 + 0 weeks	10 (43.5)	28 (60.9)	47 (51.6)
>24 + 0 weeks	7 (30.4)	12 (26.1)	21 (23.1)
Termination of pregnancy (TOP) ^1^	16 (69.6)	32 (69.6)	50 (54.9)
TOP with Feticide ^1^	2/16 (12.5)	2/32 (6.3)	11/50 (22.0)
Intention to continue the pregnancy	7 (30.4)	14 (30.4)	41 (45.1)
Death in utero or during birth ^1^	2/7 (28.6)	9/14 (64.3)	5/41 (12.2)
Livebirths ^1^	5/7 (71.4)	5/14 (35.7)	36/41 (87.8)

^1^ number (percent), ^2^ median (interquartile range), ^3^ mean (standard deviation). CHD, congenital heart disease.

**Table 2 life-12-01223-t002:** Cardiovascular anomalies among 160 fetuses with congenital heart disease (CHD) and common trisomies.

CHD	T13 (*n* = 23)	T18 (*n* = 46)	T21 (*n* = 91)
**Septation defects**			
Isolated ventricular septal defect	1 (4.3)	19 (41.3)	16 (17.6)
Common atrium	1 (4.3)	3 (6.5)	7 (7.7)
Atrio-ventricular septal defect	2 (8.7)	9 (19.6)	56 (61.5)
**Anomalies of the ventricles**			
Hypoplastic left heart syndrome	2 (8.7)	6 (13.0)	2 (2.2)
Hypoplastic right heart syndrome	1 (4.3)	1 (2.2)	1 (1.1)
**Ventriculo-arterial anomalies**			
d-Transposition of the great arteries	1 (4.3)	0	1 (1.1)
Tetralogy of Fallot	4 (17.4)	3 (6.5)	9 (9.9)
Double-outlet right ventricle	4 (17.4)	11 (23.9)	4 (4.4)
Truncus arteriosus communis	1 (4.3)	0	0
**Outflow tract anomalies**			
Aortic stenosis/atresia	0	1 (2.2)	1 (1.1)
Coarctation of the aorta	1 (4.3)	14 (30.4)	15 (16.5)
Right-sided aortic arch	3 (13.0)	0	3 (3.3)
Aberrant right subclavian artery	1 (4.3)	0	1 (1.1)
Pulmonary stenosis/atresia	0	1 (2.2)	6 (6.6)
Mitral stenosis/atresia	3 (13.0)	3 (6.5)	1 (1.1)
**Complex CHD**	7 (30.4)	2 (4.4)	0
**CHD not further specified**	0	1 (2.2)	6 (6.6)
**Fetal bradycardia**	0	0	1 (1.1)
**Isolated pericardial effusion**	0	0	1 (1.1)

Data are presented as number (percent). Some fetuses had more than one cardiac anomaly. *CHD*, congenital heart disease.

**Table 3 life-12-01223-t003:** Extracardiac Anomalies and Sonographic Soft Markers.

Anomaly	T13 (*n* = 23)	T18 (*n* = 46)	T21 (*n* = 91)
**Extracardiac anomalies (total)**	22 (95.7)	44 (95.7)	31 (34.1)
Nervous system, spina bifida	12 (52.2)	23 (50.0)	5 (5.5)
Eye, ear, face	15 (65.2)	18 (39.1)	1 (1.1)
Digestive system	0	2 (4.3)	8 (8.8)
Abdominal wall defects	7 (30.4)	9 (19.6)	0
Urinary	5 (21.7)	5 (10.9)	4 (4.4)
Limb	12 (52.2)	27 (58.7)	5 (5.5)
Other anomalies	3 (13.0)	6 (13.0)	13 (14.3)
**Sonographic soft markers (total)**	8 (34.8)	29 (63.0)	27 (29.7)
Single umbilical artery	6 (26.1)	15 (32.6)	2 (2.2)
Unfused amnion and chorion	1 (4.3)	3 (6.5)	13 (14.3)
Plexus choroideus cyst(s)	0	12 (26.1)	0
Absent/hypoplastic nasal bone	0	3 (6.5)	11 (12.1)
Polyhydramnios	0	5 (10.9)	3 (3.3)
Hyperechogenic bowel	0	1 (2.2)	0

Data are presented as number (percent). Some fetuses had more than one extracardiac anomaly or sonographic soft marker.

**Table 4 life-12-01223-t004:** Maternal morbidity: gestational and perinatal complications of pregnancy and delivery in women with fetal congenital heart disease and trisomy 13, 18, or 21.

Maternal Morbidity	T13 (*n* = 7)	T18 (*n* = 14)	T21 (*n* = 41)
Gestational diabetes mellitus	0	1 (7.1)	5 (12.2)
Hypertensive disorders of pregnancy	2 (28.6)	1 (7.1)	3 (7.3)
Premature rupture of membranes	0	2 (14.3)	1 (2.4)
Peripartum bleeding *	0	2 (14.3)	2 (4.9)
Emergency caesarean section	0	0	1 (2.4)
Curettage	0	0	2 (4.9)

Data are presented as number (percent); some women had more than one complication. Only cases with the intention to continue the pregnancies were included. * blood loss >500 mL for vaginal deliveries, >1000 mL for caesarean section.

**Table 5 life-12-01223-t005:** Birth characteristics of livebirths.

Variable	T13 (*n* = 5)	T18 (*n* = 5)	T21 (*n* = 36)
Gestational age at delivery (weeks) ^2^	36.3 (36.1–38.4)	36.1 (34.0–40.0)	37.4 (35.7–38.6)
Preterm delivery (≤37.0 weeks) ^1^	3 (60)	3 (60)	14 (38.9)
Delivery mode caesarean ^1^	2 (40)	2 (40)	20 (55.6)
Birth weight (grams) ^3^	2036.6 (557.6)	1668.0(843.8)	2581.7 (819.4)
Birth length (centimeter) ^2^	45.6 (5.5)	39.0 (7.0)	46.9 (5.3)
APGAR 1 min ^2^	4.0 (2.0–5.0)	7.5 (4.0–8.5)	8.0 (7.0–9.0)
APGAR 5 min ^2^	5.0 (2.0–6.0)	8.0 (4.0–9.5)	9.0 (8.0–10.0)
APGAR 10 min ^2^	5.0 (4.0–7.0)	9.5 (5.0–10.0)	9.5 (9.0–10.0)
Umbilical cord pH ^2^	7.2 (7.2–7.2)	7.4 (7.4–7.4)	7.3 (7.3–7.3)
Early neonatal deaths ^1^	5 (100)	3 (60)	1 (2.8)
Neonatal survival ^1^	0	0	31 (86.1)

Only cases with the intention to continue the pregnancy and with liveborn neonates were included. ^1^ number (percent), ^2^ median (interquartile range), ^3^ mean (standard deviation).

## Data Availability

The data presented in this study are available on request from the corresponding author.

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
