# Peer review of "Outcome after Prenatal Diagnosis of Trisomy 13, 18, and 21 in Fetuses with Congenital Heart Disease"

_life, 2022, doi:10.3390/life12081223_

Round 1
Reviewer 1 Report
In the article under the title: “Outcome after prenatal diagnosis of trisomy 13, 18, and 21 in fetuses with congenital heart disease” the authors describe the pregnancy outcomes in the presence of fetal congenital heart diseases and common trisomies from a single tertiary referral center during 2000-2020 in a retrospective cohort study. The primary aim was to assess the stillbirth and early and late neonatal mortality rates for pregnancies complicated by common chromosomal aneuploidies (trisomies 13, 18, and 21) in the presence of CHD. Secondary aims were to investigate maternal morbidity in these pregnancies, when the parents had decided to continue the pregnancy in the presence of fetal CHD and aneuploidy, and to study the gestational or neonatal age when mortality occurred. The data presented herein and in other similar studies may be helpful for counseling affected parents when the decision to terminate or continue the pregnancy should be considered.
The entire article is well written, illustrated, and accompanied by an appropriate reference list.
However, some obstacles must be removed before the article is accepted for publication.
These, among others, include:
The authors have written: Extracardiac anomalies (ECA) were diagnosed prenatally in 100 (62.5%) pregnancies. However, in table 3, the total number of ECA is 22 +44+31=97. Please correct or explain.
The authors have written: Additionally, in 54 (33.8%) cases, sonographic soft markers were found (trisomy 13: 8 fetuses (34.8%), trisomy 18: 29 fetuses (70.7%) and trisomy 21: 27 fetuses (29.7%)).- These numbers are different from the ones presented in table 3. For example, in table 3, it is written for the trisomy 18: 29 fetuses (70.7%), not 29 fetuses (70.7%). Please correct or explain.
There are no tabular data for the univariate and multivariate analysis described in lines 155-165. Please provide the tabular data or add the notification – tabular information not presented.
The authors have written: IUFD/death during delivery occurred in2 (28.6%) fetuses with trisomy 13, 9 (69.2%) with trisomy 18, and 5 (12.2%) in fetuses with trisomy 21 (Table 1). However, the numerical data presented in table 1 are different from those stated in the text [2 (8.7) 9 (19.6) 5 (5.5)]. Please correct or explain.
The careful reexamination of numerical data presented in the tables and manuscript text is recommended.
A minor -to-major revision of the manuscript is recommended.
Author Response
Thank you for your important comments. Please find attached the point-by-point answer.

Reviewer 2 Report
Manuscript titled 'Outcome after prenatal diagnosis of trisomy 13, 18, and 21 in fetuses with congenital heart disease' presents retrospective study from a single tertiary referral center. The aim of this study is to provide additional data for counseling affected parents when the decision to terminate or continue the pregnancy should be considered.
Manuscript is clear and presented in a well-structured manner and at the appropriate level of English language. It fits to the scope of Life (biomedicine).
Results are presented in a several tables which are easy to read, understand and interpret. Manuscript also provides data about type of cardiovascular anomalies presented and extracardiac anomalies and sonographic soft markers.
Although trisomy 13, 18 and 21 are common known chromosomopathys, and one might say that there is no novelty in this subject, I still find it important to publish presented data. For the last 11 years, noninvasive prenatal testing (NIPT) for T21, T18 and T13 has become validated, widely available and performed, revealing cases of trisomy even earlier in pregnancy compared to time before NIPT. So even more future parents will seek genetic counselling. The value of this manuscript is that it presents the stillbirth and mortality rates for pregnancies complicated by chromosomopathy in the presence of CHD from the rather large studied cohort. It also investigated maternal morbidity in these pregnancies and presents data regarding gestational or neonatal age when mortality occurred. Majority of cases are followed in the same center and the number or lost to follow-up was small.
The manuscript will not address wide audience, but could be very interested to a certain population. Since manuscript does not require any additional corrections, I recommend it to be published in a current form.
Author Response
Thank you very much for acknowledging our work. We are very pleased about your comments.
Reviewer 3 Report
In general, the authors have performed a well written manuscript about an important aspect of congenital heart disease. Current evaluation of pregnant women has significantly improved the detection of these birth deficits (trisomies 13, 18, 21), which are associated with fetal congenital heart disease and increased mortality and morbidity. My comments for the authors are the following:
Major comments
1. It is crucial to present in the methods section information about the informed consent and the ethical approval of the study from the Ethics Committee/IRB in a clear and organized way.
2. You should present in the discussion the recommendations/guidelines of medical societies of Gynaecology about termination/management of pregnancies with trisomies and congenital heart disease problems as well as parent information strategies and neonate support.
Minor comments
1. Please, present less details in the abstract in order to make it more interesting for the audience.
1. Also, clarify in the abstract which are the inclusion criteria and the aims (primary and secondary) of your study.
Author Response
Thank you for your important comments. Please see the attachment.

Reviewer 4 Report
Nowadays, the availability of non-invasive prenatal test has led to early diagnosis of chromosomal anomalies in pregnancy. In the “decision-making” for parents, is useful the knowledge about prenatal, perinatal, and postnatal outcomes, and the feasibility to medical care use for fetuses and infants. Recent evidence suggests that there is variability in outcomes that may be improved by postnatal interventions, and that quality-of-life assumptions are subjective. Physicians rarely understand the criteria that family use in determining their perception or what the infant might experience. Based on results here reported, only infants with trisomy 21 survived the neonatal period. However, there is a recent paper of Cortezzo (Cortezzo DE, Tolusso LK, Swarr DT. Perinatal Outcomes of Fetuses and Infants Diagnosed with Trisomy 13 or Trisomy 18. J Pediatr. 2022 Apr 19:S0022-3476(22)00324-9. doi: 10.1016/j.jpeds.2022.04.010. Epub ahead of print. PMID: 35452657) demonstrating that, although many infants with T13 or T18 did not survive past the first week of life, nearly 20% lived for more than 1 year with varying degrees of medical support. Probably, 160 cases represent a population too small to drive a so strong conclusion
Author Response
Thank you for your important comments. Please see attachment

Round 2
Reviewer 4 Report
The manuscript has been improved after this revision and it is suitable for publication